# Hsp100 Molecular Chaperone ClpB and Its Role in Virulence of Bacterial Pathogens

**DOI:** 10.3390/ijms22105319

**Published:** 2021-05-18

**Authors:** Sabina Kędzierska-Mieszkowska, Michal Zolkiewski

**Affiliations:** 1Department of General and Medical Biochemistry, Faculty of Biology, University of Gdańsk, 80-308 Gdańsk, Poland; 2Department of Biochemistry and Molecular Biophysics, Kansas State University, Manhattan, KS 66506, USA; michalz@ksu.edu

**Keywords:** bacteria, ClpB, human, infection, molecular chaperone, pathogen, virulence

## Abstract

This review focuses on the molecular chaperone ClpB that belongs to the Hsp100/Clp subfamily of the AAA+ ATPases and its biological function in selected bacterial pathogens, causing a variety of human infectious diseases, including zoonoses. It has been established that ClpB disaggregates and reactivates aggregated cellular proteins. It has been postulated that ClpB’s protein disaggregation activity supports the survival of pathogenic bacteria under host-induced stresses (e.g., high temperature and oxidative stress), which allows them to rapidly adapt to the human host and establish infection. Interestingly, ClpB may also perform other functions in pathogenic bacteria, which are required for their virulence. Since ClpB is not found in human cells, this chaperone emerges as an attractive target for novel antimicrobial therapies in combating bacterial infections.

## 1. Introduction

Bacteria are one of the epidemiological agents that can cause infectious diseases in humans. It is estimated that approximately 10% of the known infectious agents are bacteria [1]. However, it should be emphasized that the vast majority of bacteria are not harmful to humans and some of them are beneficial, such as intestinal bacteria which facilitate food digestion in the human gastrointestinal tract and are involved in the production of vitamins necessary for human physiology. Besides, bacteria play an important role in maintaining the ecological balance in the environment they occupy. Thus, only a fraction of bacteria is pathogenic, and causes diseases in humans and animals. A disease manifestation is a consequence of the bacteria prevailing over the defenses of a mammalian immune system. Diseases caused by pathogenic bacteria are as diverse as the bacteria themselves and include pneumonia, food- and water-borne infections, wound and bloodstream infections (known as septicemia or sepsis), or sexually transmitted diseases, such as gonorrhea and chlamydiosis. Bacterial infectious diseases have a very significant impact on public health [2]. Notably, lower respiratory infections, infectious diarrhea, and tuberculosis, caused by bacterial pathogens, are among the top causes of human mortality in the world [3]. Some bacteria are responsible for zoonotic infections which can be transmitted from animals to humans via direct or indirect contacts, sometimes also by an invertebrate vector. Importantly, bacterial zoonoses cause millions of human deaths every year, which has a significant impact on global public health and economies worldwide [4]. Bacterial zoonoses also cause major losses in livestock and impact the farming industry [5]. Farm animals are often reservoirs of zoonotic pathogens [4,6]. Therefore, a “One Health” approach to both medical and veterinary care, promoted by the U.S. Centers for Disease Control and Prevention, is essential for controlling the spread of zoonotic bacterial diseases.

Understanding how bacterial infections spread and how pathogenic bacteria outsmart the host’s defenses is crucial to controlling all bacterial diseases. During the infection process, an intense battle for survival takes place between an invading pathogen and the human body. The outcome of this battle depends on responses from both the pathogen and its host. In response to a bacterial intruder, a human host activates its defense and protective mechanisms, including the innate and the adaptive immune systems, which play a critical role in protection against a broad variety of pathogens, including bacteria. These two systems are sequentially activated during infection and work together to fight the invaders and destroy the initial infection [7]. In turn, pathogenic bacteria have evolved diverse strategies of interacting with the human host and rapid adaptation to new environmental conditions, including elevated temperature or oxidative stress, which are associated with the host’s immune responses. Thanks to these strategies, a bacterial intruder can establish itself within an infected host, i.e., it can survive, replicate and spread inside the human body, leading to a disease manifestation. Pathogenic bacteria use a number of virulence mechanisms and factors as effective weapons in subverting the human host and turning the host’s cellular processes to their advantage.

The molecular chaperone ClpB which belongs to the Hsp100/Clp subfamily of the AAA+ ATPases (ATPases associated with diverse cellular activities) is one of the factors which can enhance bacterial survival in the host during infection. Hsp100 chaperones are present in bacteria, protozoa, fungi, and plants, but not in animals and humans. The aim of this review is to discuss the current understanding of the role of ClpB in selected pathogenic bacteria and its importance during bacterial infections in humans. Importantly, since ClpB is not found in human cells, this chaperone becomes a promising target for novel antibacterial strategies, which is particularly important nowadays due to an increasing rate of developing antibiotic resistance among pathogenic bacteria, an alarming problem for global public health [8,9].

## 2. AAA+ Chaperone ClpB: Its Structure, Function and Mechanism of Action

Bacterial ClpB and its yeast homolog, Hsp104 are the best-studied members of the Hsp100 chaperone family [10,11]. ClpB is found in the vast majority of bacteria with the exception of distinct bacterial species, such as the Gram positive *Bacillus subtilis* [12,13]. This year marks three decades since ClpB was first identified as a heat shock protein necessary for the optimal growth and survival of *E. coli* at high temperature and thus required for the thermotolerance of this bacterium [14]. Unlike many other Clp family members that form complexes with peptidase subunits and participate in protein degradation, ClpB cooperates with the DnaK chaperone system to solubilize and reactivate aggregated proteins accumulating in bacteria under stress conditions [15,16,17,18]. ClpB can remodel some protein substrates without the DnaK assistance, but cooperation of ClpB and DnaK produces the most efficient disaggregation [19,20,21]. Interestingly, the ClpB-DnaK cooperation in protein disaggregation is species-specific, i.e., ClpB efficiently cooperates only with DnaK from the same microorganism [22,23,24]. 

Like other Hsp100 family members, ClpB forms ring-shaped hexamers in the presence of ATP [25,26,27,28] with a narrow central channel (pore), wide enough to accommodate extended unfolded polypeptides (Figure 1). Each protomer of ClpB consists of multiple domains: N-terminal domain (NTD), two ATP-binding modules, i.e., nucleotide-binding domain 1 (NBD-1) and nucleotide-binding domain 2 (NBD-2) and a unique coiled-coil middle domain (MD) inserted at the end of NBD-1 (Figure 1A). Each NBD domain contains all the characteristic and highly conserved motifs of AAA+ ATPases, namely Walker A, Walker B, arginine finger, sensor-1, and sensor-2 (Figure 1A). The NTD of ClpB is responsible for the recognition and binding of protein substrates. NBDs generate energy from ATP for polypeptide translocation through the ClpB channel. The MD mediates the interactions with DnaK that are required for bacterial thermotolerance and efficient protein disaggregation [23,24,29] and is also involved in coordinating the communication between NBDs [19]. Of note, the presence of the coiled-coil MD distinguishes ClpB from such Hsp100/Clp proteins as ClpA, ClpX or HslU that are associated with a peptidase (ClpP or HslV) and do not display disaggregase activity. 

Elegant studies of Bukau’s group demonstrated that protein disaggregation mediated by ClpB is linked to the ATP hydrolysis-coupled substrate translocation through the central channel [31]. Recent advances in high-resolution cryo-electron microscopy and single-molecule force spectroscopy provided critical insights into the mechanism of ClpB activity. Most significantly, recent cryo-EM image reconstructions revealed that the subunits of hexameric ClpB/Hsp104 (Figure 1B) are arranged in a spiral configuration and undergo dynamic conformational rearrangements which support ratcheting of substrates through the central channel [30,32,33,34]. Optical tweezer experiments demonstrated that ClpB is a powerful source of a mechanical force capable of extracting polypeptides from aggregated particles and possibly acting upon surface-exposed loops (Figure 2) [35]. In contrast to a directional force generation by ClpB, DnaK and other Hsp70 chaperones modulate the conformation of their substrates by applying “entropic pulling” and stochastic interactions [36,37].

In summary, ClpB is a pivotal component of protein quality control which maintains protein homeostasis (proteostasis) in bacterial cells and supports their survival under environmental stresses by mediating the reactivation of protein aggregates. Due to their unique protein disaggregation activity, ClpB and its yeast ortholog Hsp104 were postulated to become tools in the development of novel therapies for human protein aggregation diseases, such as Alzheimer’s disease, Parkinson’s disease, or Huntington’s disease. However, no ClpB/Hsp104 orthologs were found in the human and animal proteomes [38], which created an opportunity to explore the potential of this chaperone as a novel antimicrobial target.

## 3. The Role of ClpB in Bacterial Pathogens

During the past few decades, a number of studies revealed the ClpB involvement in the virulence of many bacterial pathogens, but a specific function of ClpB during infection remains to be fully elucidated. Below, we discuss the role of ClpB in the group of bacterial pathogens where that chaperone’s function was investigated (Table 1). Included among those pathogens are both Gram-positive and Gram-negative bacteria, gastrointestinal pathogens that must contend with diverse physical and chemical stresses, and also bacteria that cause animal-to-human zoonotic infections, such as leptospirosis, ehrlichiosis or fmia. Among the bacteria discussed below, *Ehrlichia chaffeeensis* and *Mycoplasma pneumoniae* are obligate intracellular pathogens that exclusively grow and replicate inside the host cells, while others are facultative pathogens, capable of proliferation both inside the host cells and in environmental niches.

In *Staphylococcus aureus, Francisella tularensis*, and *Porphyromonas gingivalis*, ClpB is required for survival under stress conditions and intracellular proliferation in in vitro and in vivo models [39,40,48,49]. Interestingly, recent experiments utilizing *F. tularensis* subspecies *holarctica* and *tularensis* demonstrated that ClpB performs multiple functions in these subspecies and its activity is important not only for the intracellular replication of these pathogens, but also for type VI secretion system (T6SS), which is essential for their virulence. Of note, T6SS in bacteria requires the assistance of ATPases, typically ClpV and IcmF. However, *F. tularensis* lacks both ClpV and the Walker A motif in IcmF (which is crucial for the ATPase activity), but nevertheless exhibits a functional T6SS. It has been demonstrated that in the absence of ClpB, T6SS in *F. tularensis* is severely impaired as compared to the wild-type strain [49]. Therefore, it was postulated that ClpB compensates for the ClpV absence in *F. tularensis*, contributes to the assembly−disassembly cycle of the T6SS apparatus, and generates the energy required for T6SS [49].

Furthermore, it has been demonstrated that the *P. gingivalis clpB null* mutant exhibits low invasiveness and attenuated virulence in a murine model of infection [50]. The virulence of *Listeria monocytogenes*, *Salmonella typhimurium* and *Mycobacterium tuberculosis clpB null* mutants was also significantly attenuated in the infection models, as compared to the wild-type and complemented strains [44,51,52]. Additionally, in the case of *M. tuberculosis*, it has been shown that ClpB is an important mediator of resistance against proteotoxic stress caused by the sequestration of IOPs (irreversibly oxidized proteins) and followed by the asymmetric distribution of aggregates within bacteria and between their progeny [52]. Thanks to such a distribution of aggregated proteins, some bacterial progeny contain a minimal amount of IOPs and cope with stressful conditions much more efficiently than their IOP-burdened siblings. Furthermore, ClpB is required for induced thermotolerance of *L. monocytogenes* and *Enterococcus faecalis* [44,53]. The *E. faecalis clpB null* strain also showed a reduced virulence in the *Galleria mellonella* infection model [53].

A loss of the ClpB function in the pathogenic spirochaete *Leptospira interrogans* resulted in bacterial growth defects under stress conditions, such as thermal and oxidative stresses and nutrient limitation [54]. Moreover, it has been shown that the ClpB deficiency made that pathogen avirulent in an animal model of acute leptospirosis (in gerbils), as compared to its parental strain [54]. The role and significance of ClpB in leptospiral virulence and pathogenesis of leptospirosis was discussed in detail in a recent review [43]. Briefly, recent results [55] indicated that the ClpB-mediated protein disaggregation activity is responsible for maintaining energy homeostasis under stress conditions in *L. interrogans* by protecting the conformational integrity and catalytic activity of metabolic enzymes, which are sensitive to stress and prone to aggregation. Thus, it could be proposed that ClpB plays an important role in the *L. interrogans* stress-induced adaptation to mammalian hosts. Notably, ClpB is one of the *L. interrogans* hub proteins interacting with the host factors, including the components of ECM (extracellular matrix) and the host plasma [56]. Furthermore, it is also likely that *L. interrogans* ClpB is involved in evading the toll-like receptor responses. Taken together, the previous results suggest that ClpB may be involved in the adhesion of *Leptospira* to the surface of host cells, plasminogen acquisition, and the immune evasion. Certainly, further studies are needed to fully resolve the role of ClpB in leptospiral virulence and the pathogenesis of leptospirosis, a zoonotic disease with a significant impact on public health worldwide.

Recently, the role of ClpB in *Mycobacterium tuberculosis* has been also investigated in more detail [13]. It has been demonstrated that ClpB is required for the survival of *M. tuberculosis* under stressful conditions and is involved in regulating its virulence. Additionally, the same study found that ClpB is necessary for the maintenance of dormant *M. tuberculosis* in latency-like conditions, such as prolonged hypoxia and nutrient starvation [13]. Such conditions exist inside granulomas that form during the host’s immune response to *M. tuberculosis*. The tuberculosis granulomas create not only an immune microenvironment for controlling the infection, but also provide a niche in which mycobacteria can survive in a dormant state [57]. Since, dormancy ensures the survival of mycobacteria inside hypoxic granulomas, ClpB may be one of the key players in maintaining the persistence of *M. tuberculosis* within its host. Interestingly, it is also possible that ClpB, together with its DnaK partner, contribute to a recovery from dormancy and restoration of mycobacterial cell activity [58], which occurs in 10% of the latently infected individuals [57]. Therefore, it is likely that ClpB also supports an escape strategy of mycobacteria from granulomas and their spread throughout the body, which results in disease progression [57]. Furthermore, it has been shown for the first time that some fraction of ClpB, a cytosolic protein, can be secreted into the extracellular environment and can interact with host macrophages. Hence, ClpB mediates the inflammatory immune response which may help in maintaining the integrity of tuberculous granulomas containing the pathogen [13]. Taken together, the studies described above have revealed that ClpB acts not only as a stress-response factor in pathogenic bacteria, but it may also function as a signaling molecule.

ClpB is also required for in-host survival of *Mycoplasma pneumoniae,* the smallest self-replicating microorganism [59]. Moreover, it was demonstrated that ClpB can elicit an immune response in experimentally infected mice and patients infected with *M. pneumoniae*, again, highlighting its role in the *M. pneumoniae* virulence. ClpB-mediated activation of the host’s immune response was also found during infections with *L. interrogans* and *F. tularensis* [60,61]. The documented secretion of *M. tuberculosis* ClpB into the extracellular environment [13] may explain an apparent immunogenicity of ClpB from pathogenic bacteria. Finally, it has been found that in *Ehrlichia chaffeensis,* during infection of mammalian cells, the *clpB* expression is elevated and its induction correlates with the pathogen’s replicating stage inside host cells, which demonstrates an essential role of ClpB in the *Ehrlichia* response to the host-induced stress [20].

A number of studies described above have collectively demonstrated an essential role of ClpB in host invasion and the rapid adaptation of bacterial pathogens to their hosts, pathogen survival and replication, and evasion of the host defense mechanisms. Thus, ClpB performs multiple functions in bacterial pathogens (Figure 3). As described above, the biochemical activity of ClpB as an energy-dependent disaggregase, has been well documented. A link between infection and the aggregation propensity of the pathogen’s proteins has only begun to emerge [62,63]. It needs to be determined if protein aggregation is a universal factor that affects the pathogens’ survival in infected hosts and how the disaggregase activity of ClpB supports the various adaptive processes that help establish bacterial infections.

## 4. ClpB as a Druggable Target

The absence of ClpB orthologs in mammals, including humans, makes this chaperone an attractive target in combating bacterial infections. As described above, ClpB is an essential factor in bacterial stress response and pathogen virulence. Thus, inhibition of ClpB might suppress infectivity and the survival of invading pathogens. However, selective and high-affinity inhibitors of ClpB are not currently available.

Studies on the Hsp104-supported propagation of yeast prions identified millimolar-range guanidinium chloride (GdmHCl) as a surprisingly effective and selective inhibitor of Hsp104 [64]. A millimolar concentration of GdmHCl is insufficient to induce protein denaturation, but it inhibits the ATPase activity of ClpB/Hsp104 in vitro and impairs thermotolerance in bacteria and yeast [15,65,66]. A crystal structure of NBD-1 from *Thermus thermophilus* ClpB in the presence of ADP and GdmHCl revealed that the guanidinium cation binds in close proximity to the nucleotide, directly above the adenine ring and the 2′-OH of ribose [67]. Biochemical studies showed the inhibitory effect of Gdm^+^ to arise from the stabilization of the ClpB-nucleotide complex and the reduction of the nucleotide dissociation rate [67].

Weak binding affinity (*K*_diss_ ≈ 1 mM) and possible off-target effects make GdmHCl unsuitable for further development as a drug candidate. However, GdmHCl became a useful tool to probe the biological function of ClpB in pathogenic bacteria. For instance, the ATPase activity of ClpB from the tick-transmitted pathogen, *Ehrlichia chaffeensis* was significantly inhibited in vitro by 1 mM GdmHCl [62]. In a culture of *Ehrlichia*-infected macrophages, pathogen viability was reduced by ~60% in the presence of 0.5 mM GdmHCl without affecting host cells. Furthermore, the size of the aggregated protein fraction in *E. chaffeensis* increased significantly in cultures supplemented with 0.5 mM GdmHCl, which also resulted in preferential accumulation of ClpB with the aggregated proteins [62], consistent with trapping of chaperone-aggregate complexes due to slow ATP turnover.

Several groups performed unbiased screens of chemical libraries in search for other ClpB inhibitor leads. A high-throughput screen for ClpB-interacting compounds [68] identified several ClpB ligands that unfortunately belong to the promiscuous “pan assay interference compounds” (“PAINS”) [69] and the remaining promising compounds exhibited off-target effects. Another screen identified suramin as a ligand for Hsp100 [70]; however, suramin is a known promiscuous inhibitor of many ATP-binding proteins [71,72]. Consequently, no ClpB inhibitor has been identified so far by unbiased screens that might represent a promising lead for drug development.

A recent study explored the hypothesis that known small-molecule ligands of AAA+ ATPases might also interact with ClpB. Among three inhibitors of the human AAA+ ATPase p97, a promising antitumor target, only one affected ClpB [73]. The identified compound, N^2^,N^4^-dibenzylquinazoline-2,4-diamine (DBeQ) has been previously used as an antimicrobial [74,75], but without clear identification of its cellular targets. It has now been shown that ClpB is the main target of DBeQ in *E. coli* under both permissive conditions and during heat stress [73]. Thus, ClpB can be selectively targeted with a small-molecule ligand in bacterial cells and such a treatment could produce a loss of bacterial viability. Interestingly, targeting ClpB with DBeQ in *E. coli* cells produced toxicity that transcended a loss-of-function phenotype observed in the *clpB*-null strain. Toxicity of ClpB in bacteria and Hsp104 in yeast has been observed before for “hyperactive” protein variants with mutations within the middle domain [76,77,78]. Whether an analogous toxic gain of function occurs in the DBeQ-treated ClpB remains to be investigated.

## 5. Conclusions

In this review, we highlighted an important role of the Hsp100 chaperone ClpB in supporting the virulence and survival of a broad range of pathogenic bacteria. Multiple studies revealed that the ClpB-mediated protein disaggregation activity plays a key role in the pathogen response to the host-induced proteotoxic stresses. The effectiveness of this response determines the pathogen’s ability to adapt to the stress, proliferate in the host cells, and induce disease symptoms. Interestingly, ClpB may also perform other functions in pathogenic bacteria, which are necessary for their virulence and may be linked to disease (see Figure 3). Importantly, the absence of ClpB orthologs in humans and its demonstrated druggability make this chaperone a high-priority target for inhibitor development.

Global spread of antibiotic resistance among bacterial pathogens (e.g., methicillin-resistant *Staphylococcus aureus* (MRSA), vancomycin-resistant *Enterococcus* (VRE) multi-drug-resistant *Mycobacterium*
*tuberculosis* (MDR-TB), carbapenem-resistant *Enterobacteriaceae* (CRE) gut bacteria) became one of the most serious public health challenges of the twenty-first century. The development of novel antibacterial strategies and the discovery of new antibiotics is crucial for meeting that challenge. Targeting microbial Hsp100 chaperones provides a promising pathway for the development of innovative therapeutic strategies in combating a broad range of infectious diseases.

## Figures and Tables

**Figure 1 ijms-22-05319-f001:**
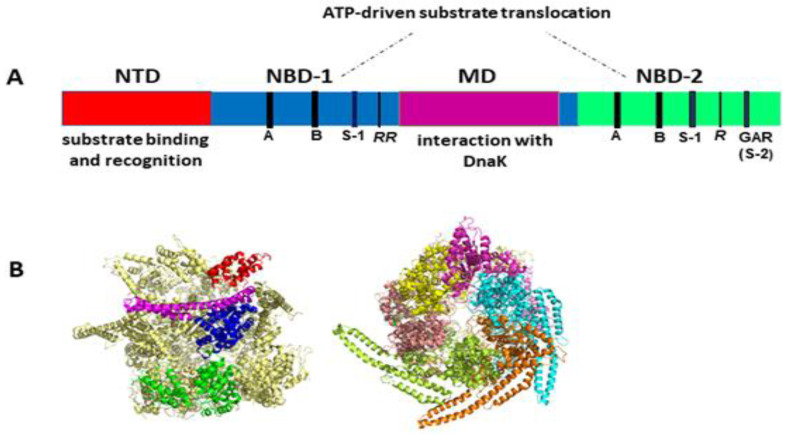
(**A**) Structural organization of the ClpB monomer. Four domains are indicated: N-terminal domain (NTD), two nucleotide-binding domains (NBD-1 and NBD-2), and middle domain (MD). Each NBD contains the characteristic AAA+ motifs: Walker A (GX_4_GKT/S) (**A**), Walker B (Hy_2_DE) (B), sensor-1 (S-1), sensor-2 (S-2, GAR), and the arginine fingers (*R*). (**B**) Cryo-EM structure of hexameric Hsp104 from *S. cerevisiae* in the closed conformation (PDB entry 6N8T) [30]. Left panel: side view with the structural domains indicated for one Hsp104 subunit: NTD (red), NBD-1 (blue), MD (magenta), and NBD-2 (green). Right panel: top view with each subunit shown in a different color. The substrate-processing channel is visible at the center of the structure. Three out of six MDs were resolved in this cryo-EM image analysis, which highlights the highly dynamic properties and structural asymmetry of the hexameric complex. Images generated using PyMol 1.3 (Schrödinger LLC, www.pymol.com accessed on 2010).

**Figure 2 ijms-22-05319-f002:**
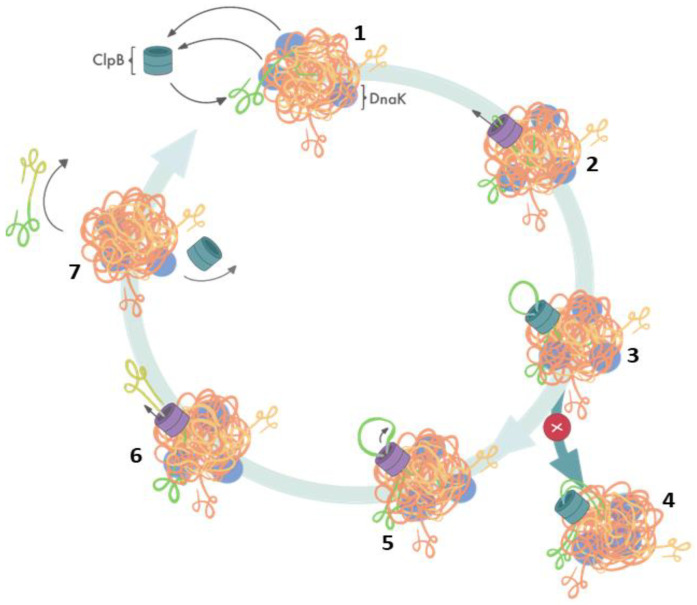
Cooperation of ClpB and DnaK during aggregate reactivation based on ref. [35]. (1) The aggregate-bound DnaK recruits ClpB to a protein aggregate and exposes ClpB-accessible fragments of the aggregate; (2) ClpB initiates substrate translocation from an exposed polypeptide loop; (3) stably folded domains can become obstacles for ClpB-mediated polypeptide extraction; (4) resistance during the translocation stalls ClpB-mediated disaggregation; (5) switching to single-strand translocation can release stalled ClpB; (6) extracted unfolded polypeptide exits the ClpB channel; (7) extracted polypeptide refolds while ClpB can engage in another polypeptide extraction cycle.

**Figure 3 ijms-22-05319-f003:**
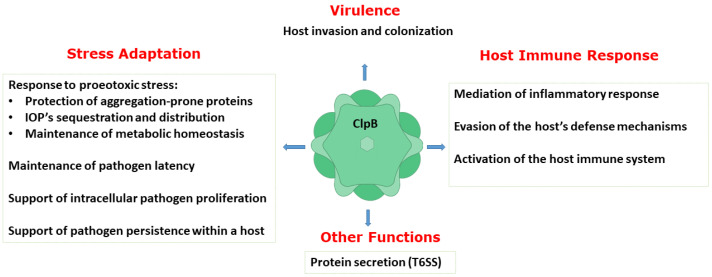
Summary of the documented functions of ClpB in supporting virulence, host−pathogen interactions, and stress adaptation in pathogenic bacteria. IOPs: irreversibly oxidized proteins; T6SS: type VI secretion system.

**Table 1 ijms-22-05319-t001:** Selected bacterial pathogens and the associated diseases in humans and animals [1,39,40,41,42,43,44,45,46,47].

Bacterial Species	Disease	Transmission
*Ehrlichia chaffeensis* * (Gram-negative *Rickettsia* bacterium)	human monocytic ehrlichiosis (HME)	zoonosis transmitted through an infected tick
*Enterococcus faecalis* (Gram-positive cocci)	bacteremia, endocarditis, intra-abdominal, pelvic and soft tissue infections, and urinary tract infections	transmission via a physical contact (person−person) or a contact with contaminated surfaces
*Francisella tularensis* (Gram-negative coccobacillus)	tularemia (also known as a rabbit fever)	zoonosis transmitted to humans in numerous ways, including ticks, deerfly bites, direct handling of infected tissues, ingestion of contaminated water or tissues, or inhalation of infective materials
*Leptospira interrogans* (Gram-negative spirochaete)	leptospirosis in mammals, including humans	zoonosis (wild and domestic animals are a main source of this pathogen) transmitted mainly through urine of infected animals, contact with a urine-contaminated environment, i.e., water or moist soil
*Listeria monocytogenes* (Gram-positive rod)	food-borne infections; listeriosis of pregnancy; neonatal listeriosis; clinical syndromes associated with listeriosis: meningoencephalitis, meningitis, septicemia, spontaneous abortions, stillbirth, premature labor, and neonatal disease	infection occurs through consumption of contaminated food (unpasteurized milk, soft cheeses, vegetables and some meat products), transmission is possible from mother to fetus and from mother to child during birth
*Mycobacterium tuberculosis* (neither Gram-positive nor Gram-negative bacterium)	tuberculosis	transmission through airborne particles called droplet nuclei generated by a sick person coughing, sneezing, shouting, or singing
*Mycoplasma pneumoniae* * (Gram-negative)	acute and chronic respiratory diseases; pneumonia	transmission through droplets generated by an infected person coughing or sneezing
*Porphyromonas gingivalis* (Gram-negative rod)	periodontal diseases, chronic periodontitis, systemic diseases, including heart disease, stroke, and diabetes mellitus as well as preterm delivery of low birth-weight infants	colonization of the mouth; transmission through saliva
*Staphylococcus aureus* (Gram-positive)	wide spectrum of infections from superficial wound infections, Staph food poisoning to life threatening septicemia and toxic shock syndrome;	skin and mucous membrane colonization; transmission through direct contact (person−person)
*Salmonella enterica* serovar *typhimurium* (Gram-negative)	food-borne illness; enteric (typhoid) fever, food poisoning in humans	zoonosis transmitted through contact with infected animals, contaminated water, and the environment; food-borne infections

* Obligate intracellular pathogens.

## Data Availability

Not applicable.

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
