# Peer review of "Hsp100 Molecular Chaperone ClpB and Its Role in Virulence of Bacterial Pathogens"

_ijms, 2021, doi:10.3390/ijms22105319_

Round 1
Reviewer 1 Report
The AAA+ protein ClpB promotes survival during severe stress conditions by reactivating aggregated proteins. ClpB can also play crucial roles in pathogenic bacteria by supporting growth and resistance towards host defense mechanisms including oxidative stress. The role of ClpB in bacterial virulence and the absence of ClpB homologs in higher eukaryotes qualify the disaggregase as potential drug target. Here, the authors summarize the current knowledge on the mechanism of the disaggregase and its diverse functions in bacterial physiology. The review is overall well written and includes latest findings. There are a couple of minor changes required (see below), which the authors are trusted to address in a revised manuscript.
Critiscism:
- The authors need to better separate the different roles of ClpB in supporting bacterial virulence. These different functions are nicely displayed in figure 3 and a respective structuring of the main text is required. The review will benefit from the addition of subheadings and extra chapters to better describe the diverse roles of the disaggregase in bacterial pathogenicity.
- Page 3, figure 1: The authors should provide a recent cryo EM structure of ClpB, illustrating the organization of the individual domains.
- Page 3, line 98: the coiled-coil MD is also present in other Hsp100 proteins and therefore does not separate ClpB from other family members. It is rather the specific sequence composition of the MD that provides specificity for the Hsp70 partner (DnaK) and its function in protein disaggregation.
- Page 3, figure 2: The role of DnaK in the disaggregation process is not obvious. The authors should use a different colour for DnaK as it is hardly visible in the cartoon. Furthermore, the cartoon does not illustrate the targeting function of DnaK but implies that ClpB binds independent of DnaK to protein aggregates. While such possibility may hold true for selected model substrates in vitro, ClpB was shown to require DnaK for aggregate binding in vivo.
- Page 4, line 125; page 7, line 224: ClpB is not an essential component of the proteostasis network, as clpBknockout cells do typically not exhibit severe phenotypes at physiological growth conditions.
- Page 5, line 152: The authors should specify that the role of ClpB in the type VI secretion system (T6SS) of Francisella sp. is unique. This role is typically executed by the Hsp100 member ClpV in most T6SS-encoding bacteria. The unique function of ClpB is likely explained by the specific composition of the Francisella T6SS, which is different from canonical T6SS.
- Page 6, line 183: The proposed role of Leptospira ClpB in adhesion to the surface of host cells implies that ClpB is cell-surface exposed or even secreted. Is there evidence for such localization? In this context ClpB should also function independent of ATP, a fact that should be discussed by the authors.
- Page 6, line 192. The authors describe the role of ClpB in maintaining the dormant state of tuberculosis cells. Is this role similar or different from the function of ClpB in allowing bacterial cells to escape from dormancy (as described by Pu et al., Mol Cell 2019)?
- Page 7, line 261: DBeQ is targeting ClpB in coli cells also under permissive conditions. How do the authors envision toxic effects of DBeQ as ClpB is not required for growth under these conditions. Is it clear that the drug is inhibiting ClpB? Or can DBeQ also lead to overactivation of ClpB, which indeed has been reported by several groups to cause severe toxicity.
Author Response
Response to Reviewer #1
General comments: The AAA+ protein ClpB promotes survival during severe stress conditions by reactivating aggregated proteins. ClpB can also play crucial roles in pathogenic bacteria by supporting growth and resistance towards host defense mechanisms including oxidative stress. The role of ClpB in bacterial virulence and the absence of ClpB homologs in higher eukaryotes qualify the disaggregase as potential drug target. Here, the authors summarize the current knowledge on the mechanism of the disaggregase and its diverse functions in bacterial physiology. The review is overall well written and includes latest findings. There are a couple of minor changes required (see below), which the authors are trusted to address in a revised manuscript.
Criticism:
- The authors need to better separate the different roles of ClpB in supporting bacterial virulence. These different functions are nicely displayed in figure 3 and a respective structuring of the main text is required. The review will benefit from the addition of subheadings and extra chapters to better describe the diverse roles of the disaggregase in bacterial pathogenicity.
Response:
First of all, we would like to thank the Reviewer for all constructive suggestions and comments that helped us to improve the manuscript.
In the case of the Section 3: The role of ClpB in bacterial pathogens, we prefer to keep the previously adopted way of describing the ClpB’s functions in supporting bacterial virulence without introducing additional levels of subheadings. The role of ClpB is described in the text according to the order of the selected bacterial species, which avoids repetitions and is consistent with the information presented in Table 1. We describe the different functions of ClpB in Figure 3 to provide a function-oriented summary in addition to species-oriented information in the main text.
- Page 3, figure 1: The authors should provide a recent cryo EM structure of ClpB, illustrating the organization of the individual domains.
Response:
As suggested by the Reviewer, a recent cryo-EM structure of Hsp104 published by Lee et al. (2019; Cell Reports 26, 29–36) was added to Figure 1 in the revised manuscript. We selected the above structure because, to our knowledge, it is the only one available in PDB which resolves the N-domain of Hsp104/ClpB. We have also made changes to the figure caption, which now reads (lanes 105-109): “(B) Cryo-EM structure of hexameric Hsp104 from S. cerevisiae in the closed conformation [30]. Left panel: side view with the structural domains indicated for one Hsp104 subunit: NTD (red), NBD-1 (blue), MD (magenta), and NBD-2 (green). Right panel: top view with each subunit shown in a different color. The substrate-processing channel is visible at the center of the structure. Three out of six MDs were resolved in this cryo-EM image analysis, which highlights the highly dynamic properties and structural asymmetry of the hexameric complex.”
- Page 3, line 98: the coiled-coil MD is also present in other Hsp100 proteins and therefore does not separate ClpB from other family members. It is rather the specific sequence composition of the MD that provides specificity for the Hsp70 partner (DnaK) and its function in protein disaggregation.
Response:
We agree with this Reviewer’s comment. We have now included specific examples of proteins where MD is not present (lines 99-100); The revised text reads: “Of note, the presence of the coiled-coil MD distinguishes ClpB from such Hsp100/Clp proteins as ClpA, ClpX or HslU that are associated with a peptidase, ClpP or HslV, and do not display a disaggregase activity.”
- Page 3, figure 2: The role of DnaK in the disaggregation process is not obvious. The authors should use a different colour for DnaK as it is hardly visible in the cartoon. Furthermore, the cartoon does not illustrate the targeting function of DnaK but implies that ClpB binds independent of DnaK to protein aggregates. While such possibility may hold true for selected model substrates in vitro, ClpB was shown to require DnaK for aggregate binding in vivo.
Response:
As suggested by the Reviewer a different colour for DnaK has been used in Figure 3 and the targeting function of DnaK has been emphasized by adding arrows pointing from DnaK to ClpB. We have also made a minor change to the figure caption, which now reads (lanes 125-126): “Cooperation of ClpB and DnaK during aggregate reactivation [based on ref. 35]. (1) The aggregate-bound DnaK recruits ClpB to a protein aggregate and exposes ClpB-accessible fragments of the aggregate;”
- Page 4, line 125; page 7, line 224: ClpB is not an essential component of the proteostasis network, as clpB knockout cells do typically not exhibit severe phenotypes at physiological growth conditions.
Response:
We agree with this Reviewer’s comment and this is why we refer to conditions of stress, not the optimal physiological conditions. We have made a minor change to the text (lines 131-133), which now reads: “In summary, ClpB is a pivotal component of the protein quality control which maintains protein homeostasis (proteostasis) in bacterial cells and supports their survival under environmental stresses by mediating reactivation of protein aggregates.”
- Page 5, line 152: The authors should specify that the role of ClpB in the type VI secretion system (T6SS) of Francisella sp. is unique. This role is typically executed by the Hsp100 member ClpV in most T6SS-encoding bacteria. The unique function of ClpB is likely explained by the specific composition of the Francisella T6SS, which is different from canonical T6SS.
Response:
As suggested by the Reviewer, to better describe the role of ClpB in type VI secretion system (T6SS) of F. tularensis we have made changes in the text (lines 160-166); the revised text reads: “Of note, T6SS in bacteria requires assistance of ATPases, typically ClpV and IcmF. However, F. tularensis lacks both ClpV and the Walker A motif in IcmF (which is crucial for the ATPase activity), but nevertheless exhibits a functional T6SS. It has been demonstrated that in the absence of ClpB, T6SS in F. tularensis is severely impaired as compared to the wild-type strain [49]. Therefore, it was postulated that ClpB compensates for the ClpV absence in F. tularensis, contributes to the assembly-disassembly cycle of the T6SS apparatus, and generates energy required for T6SS [49].”
- Page 6, line 183: The proposed role of Leptospira ClpB in adhesion to the surface of host cells implies that ClpB is cell-surface exposed or even secreted. Is there evidence for such localization? In this context ClpB should also function independent of ATP, a fact that should be discussed by the authors.
Response:
Unfortunately, localization of ClpB in Leptospira has not been studied so far. The role of ClpB in adhesion of Leptospira to the host cells is proposed on the basis of the result published by Kumar and colleagues. The presence of components of extracellular matrix (ECM) among human interactors of ClpB indicates ClpB’s participation in adhesion of Leptospira. We can only speculate that a fraction of the Leptospira ClpB, in analogy to ClpB from M. tuberculosis, is secreted into extracellular space. A definite verification of this hypothesis requires further research and so does a notion of an ATP-independent function of ClpB in extracellular environment.
- Page 6, line 192. The authors describe the role of ClpB in maintaining the dormant state of tuberculosis cells. Is this role similar or different from the function of ClpB in allowing bacterial cells to escape from dormancy (as described by Pu et al., Mol Cell 2019)?
Response:
In our opinion, the role of ClpB in maintaining the dormant state of M. tuberculosis may be more complex and not limited to its disaggregase activity. The influence of ClpB on the host immune system should also be taken into account. In this case ClpB might also function as a signaling molecule. Again, a definite answer to this possibility requires further research.
An additional function of ClpB mentioned by the Reviewer has been included in the revised manuscript (lines 204-219); the revised text reads:” Such conditions exist inside granulomas that form during the host immune response to M. tuberculosis. The tuberculosis granulomas create not only an immune microenvironment for controlling the infection, but also provide a niche in which mycobacteria can survive in a dormant state [57]. Since, dormancy ensures survival of mycobacteria inside hypoxic granulomas, ClpB may be one of the key players in maintaining the persistence of M. tuberculosis within its host. Interestingly, it is also possible that ClpB together with its DnaK partner contribute to a recovery from dormancy and restoration of mycobacterial cell activity [58], which occurs in 10% of the latently infected individuals [57]. Therefore, it is likely that ClpB also supports an escape strategy of mycobacteria from granulomas and their spread throughout the body, which results in a disease progression [57]. Furthermore, it has been shown for the first time that some fraction of ClpB, a cytosolic protein, can be secreted into extracellular environment and can interact with host macrophages. Hence, ClpB mediates the inflammatory immune response which may help in maintaining the integrity of tuberculous granulomas containing the pathogen [13]. Taken together, the studies described above have revealed that ClpB acts not only as a stress-response factor in pathogenic bacteria , but it may also function as a signaling molecule.“
- Page 7, line 261: DBeQ is targeting ClpB in coli cells also under permissive conditions. How do the authors envision toxic effects of DBeQ as ClpB is not required for growth under these conditions. Is it clear that the drug is inhibiting ClpB? Or can DBeQ also lead to overactivation of ClpB, which indeed has been reported by several groups to cause severe toxicity.
Response:
We appreciate this important comment from the Reviewer. The mechanism of an apparent toxic gain of function of DBeQ-targeted ClpB in E. coli is not understood, but it is clear that the DBeQ effects are supressed in the clpB-null strain, which indicates a significant selectivity of the compound. The ClpB ATPase does not become overactive in the presence of DBeQ, as it was observed in the toxic variants with MD mutations. We added the following in the text (lanes 288-292): “Interestingly, targeting ClpB with DBeQ in E. coli cells produces toxicity that transcends a loss-of-function phenotype observed in the clpB-null strain. Toxicity of ClpB in bacteria and Hsp104 in yeast has been observed before for “hyperactive” protein variants with mutations within the middle domain [76-78]. Whether an analogous toxic gain of function occurs in the DBeQ-treated ClpB remains to be investigated.”

Reviewer 2 Report
The following corrections should be made in the manuscript.
Page 3, line 103. sensor -1
Correction: sensor-1
Page 4, line 145. [1, 38-46]
Correction: [1,38-46]
Page 6, line 219. Figure 3. Figure 3. Summary…
Correction: Figure 3. Summary…
Page 7, line 237. (Kd~1 mM)
Correction: (Kdiss ≈ 1 mM)
Page 10, line 418, Ref. 63. Chaperon. 2012,
Correction: Chaperones 2012,
Author Response
Response to Reviewer #2
Comments and suggestions: The following corrections should be made in the manuscript:
Page 3, line 103. sensor -1
Correction: sensor-1
Page 4, line 145. [1, 38-46]
Correction: [1,38-46]
Page 6, line 219. Figure 3. Figure 3. Summary…
Correction: Figure 3. Summary…
Page 7, line 237. (Kd~1 mM)
Correction: (Kdiss ≈ 1 mM)
Page 10, line 418, Ref. 63. Chaperon. 2012,
Correction: Chaperones 2012,
Response:
All the Reviewer’s suggestions have been taken into account and all the above-mentioned errors have been corrected in the revised manuscript. We would like to thank the Reviewer for his/her positive review.
